# Incomplete paralog compensation generates selective dependency on *TRA2A* in cancer

Amanda R. Lee[1,2,3], Anna Tangiyan[3], Isha Singh[3], Peter S. Choi[1,2,3]*

**1** Department of Pathology & Laboratory Medicine, University of Pennsylvania Perelman School of Medicine, Philadelphia, Pennsylvania, United States of America, **2** Cell and Molecular Biology Graduate Group, University of Pennsylvania Perelman School of Medicine, Philadelphia, Pennsylvania, United States of America, **3** Division of Cancer Pathobiology, The Children's Hospital of Philadelphia, Philadelphia, Pennsylvania, United States of America

\* peter.choi@pennmedicine.upenn.edu

## Abstract

Paralogs often exhibit functional redundancy, allowing them to effectively compensate for each other's loss. However, this buffering mechanism is frequently disrupted in cancer, exposing unique paralog-specific vulnerabilities. Here, we identify a selective dependency on the splicing factor *TRA2A*. We find that *TRA2A* and its paralog *TRA2B* are synthetic lethal partners that function as widespread and largely redundant activators of both alternative and constitutive splicing. While loss of *TRA2A* alone is typically neutral due to compensation by *TRA2B*, we discover that a subset of cancer cell lines are highly *TRA2A*-dependent. Upon *TRA2A* depletion, these cell lines exhibit a lack of paralog buffering specifically on shared splicing targets, leading to defects in mitosis and cell death. Notably, *TRA2B* overexpression rescues both the aberrant splicing and lethality associated with *TRA2A* loss, indicating that paralog compensation is dosage-sensitive. Together, these findings reveal a complex dosage-dependent relationship between paralogous splicing factors, and highlight how dysfunctional paralog buffering can create a selective dependency in cancer.

## Author summary

Cells have built-in backup systems that help them survive when key functions are disrupted. One such system involves similar genes called paralogs that can compensate for each other. This redundancy is especially important in cancer, where cells frequently acquire mutations and lose entire genes but manage to survive and proliferate. In our study, we examined two paralogs involved in RNA splicing, *TRA2A* and *TRA2B*, which help cells process genetic information correctly. We discovered that a subset of cancer cell lines are unexpectedly vulnerable to loss of just *TRA2A*, despite having its partner *TRA2B* present. In

**Data availability statement:** Data is available in the Supplementary Files and at GEO Series Accession Number GSE294883 (https://www.ncbi.nlm.nih.gov/geo/query/acc.cgi?acc=GSE294883).

**Funding:** This work was supported by National Institutes of Health grant DP2GM146251 (to P.S.C.). The funders had no role in study design, data collection and analysis, decision to publish, or preparation of the manuscript.

**Competing interests:** The authors have declared that no competing interests exist.

these sensitive cells, the natural backup system fails - *TRA2B* cannot properly maintain accurate splicing of genes essential for cell division, ultimately leading to cell death. Remarkably, providing these cells with extra *TRA2B* rescued both the splicing defects and cell survival, revealing that this backup system is dosage-sensitive. Our findings show how defective paralog compensation can create unique vulnerabilities in cancer cells, opening up possibilities for new cancer treatments that target such gene pair interactions.

## Introduction

Paralogs arise from duplication of genes followed by differing degrees of subsequent functional divergence. Due to this shared evolutionary history, they are especially enriched for genetic interactions. For example, paralogs frequently retain overlapping functions and this redundancy is thought to confer genetic robustness, allowing cells and organisms to tolerate mutational perturbations that would otherwise be inviable [1–3]. Similarly, paralog redundancy also provides cancer cells with the same protective benefit, allowing them to withstand frequent copy-number loss and ongoing genomic instability without compromising essential cellular functions [4–6]. However, when loss of a paralog does occur, whether via a positively selected or collateral event, the remaining unaltered paralog can become essential and expose a unique cancer-specific vulnerability [7]. Thus, paralogs often exemplify the phenomenon of synthetic lethality, in which the simultaneous disruption of two genes is lethal while the loss of either gene alone is tolerated [8–10].

Many such paralog-based synthetic lethalities have been discovered to date and their identification remains an attractive strategy for nominating potential therapeutic targets in cancer [11]. Well-characterized examples include *ARID1A/ARID1B* [12], *STAG1/STAG2* [13,14], *SMARCA2/SMARCA4* [15–17] and *MAGOHB/MAGOH* [18], in addition to several others [19–22]. As buffering between paralogs masks their co-essentiality in monogenic screens, recent efforts have developed a variety of multiplexed screening approaches to directly interrogate pairs of paralogs at higher-throughput [23–28]. These studies have revealed a plethora of additional synthetic lethal paralogs, with likely more remaining to be uncovered.

The process of RNA splicing is regulated by a complex network of splicing factors, many of which are paralogs [29]. This includes transformer-2 alpha and beta (*TRA2A/TRA2B*), a pair of closely-related genes within the serine/arginine-rich (SR) protein family of splicing factors [30,31]. *TRA2A* and *TRA2B* arose as paralogs early in vertebrate evolution from a single ancestral *Tra2* gene, initially identified in Drosophila where it governs splicing of female-specific isoforms during sex differentiation [32]. Like other SR proteins, TRA2A and TRA2B act primarily as splicing activators through recognition of exonic enhancer elements in their target RNAs [33–36]. Prior work has also shown that TRA2A and TRA2B share a high degree of functional overlap and, in addition to regulating alternative splicing, are involved in splicing of constitutive exons as well [37].

Here, we identify *TRA2A* as a strong and selective dependency in a diverse subset of cancer cell lines. To understand the nature of this dependency, we investigate the functional relationship between *TRA2A* and its paralog *TRA2B*, and their complementary roles in splicing regulation. We show that TRA2A and TRA2B are indeed highly redundant splicing activators that normally compensate for each other to maintain essential splicing programs. However, in select contexts, we find that this paralog-based compensatory mechanism is compromised, rendering cells vulnerable to loss of *TRA2A* alone.

## Results

### *TRA2A* is a selective dependency in a subset of cancer cell lines

Large-scale genetic screens have revealed distinct classes of cancer dependencies, with some genes showing 'outlier' effects and selective essentiality in only a subset of cancer cell lines [38]. Examining genome-wide CRISPR screening data from the Cancer Dependency Map (DepMap, 24Q2 release) [39], we found that the splicing factor *TRA2A* exhibited a pattern consistent with other selective dependencies. While *TRA2A* was largely dispensable in the majority of cell lines (avg. gene effect of -0.07), a subset of cancer lines were highly sensitive to *TRA2A* loss (n = 23 lines with gene effect < -0.5) (Fig 1A). Interestingly, TRA2A dependency was not specific to a particular cell or tumor type and was associated with a diverse range of cancer contexts (S1A Fig). We also examined DepMap data for the closely related paralog *TRA2B* and other members of the SR protein family (*SRSF1–12*) (S1B Fig). Compared to *TRA2A*, loss of *TRA2B* was more broadly detrimental across cell lines (avg. gene effect of -0.56), although it was not as essential as other SR protein family members (S1B Fig). There was also no significant correlation between the dependencies on *TRA2A* and *TRA2B* (S1J Fig).

To individually validate *TRA2A* dependency, we selected a panel of *TRA2A*-dependent cell lines (NCI-H23, NCI-H2286, LN319, PANC0504) and tumor type-matched *TRA2A*-independent cell lines (A549, LN229, AsPC-1). Using three separate sgRNAs (two sgRNAs from the DepMap library and one sgRNA independently designed), we targeted *TRA2A* across our cell line panel and measured the fitness effects over time using a competition assay (Figs 1B–1D and S1C). Consistent with the results from DepMap, we found that *TRA2A*-dependent cells were depleted following *TRA2A* knockout (KO) while *TRA2A*-independent cells were unaffected. The rate of depletion upon *TRA2A* KO varied from cell line to cell line, but in some cases was similar to or more rapid than KO of the positive control pan-essential gene *POLR2D* (S1C Fig). To confirm that sensitivity to *TRA2A* KO was an on-target effect, we expressed guide-resistant *TRA2A* cDNA (Fig 1E) and observed full rescue in competition assays across multiple *TRA2A*-dependent cell lines (Fig 1F). Finally, to validate *TRA2A*-dependency using an orthogonal approach, we utilized CRISPR interference (CRISPRi) to knockdown (KD) *TRA2A* expression (Fig 1G). Repression of *TRA2A* by KD led to a fitness defect in only *TRA2A*-dependent lines, matching our results from *TRA2A* KO (Fig 1H).

We next investigated potential predictors of *TRA2A* dependency. Data from DepMap indicated that *TRA2A* and *TRA2B* mRNA expression were similar between *TRA2A*-dependent and independent lines, with both paralogs widely co-expressed across the majority of cell lines (S1E and S1F Fig). Several *TRA2A*-dependent lines had *TRA2B* mRNA expression on the lower end of the distribution, but this was not sufficient to predict *TRA2A* dependency as other lines with comparable or lower *TRA2B* expression were *TRA2A*-independent (S1F Fig). Next, we assessed protein abundance using proteomics data from DepMap [40] and performed quantitative western blots for TRA2A and TRA2B using total protein normalization. Levels of TRA2A or TRA2B protein varied across cell lines but were comparable between *TRA2A*-dependent and independent groups, with no statistically significant differences (Figs 1I, S1D, S1H and S1I). While neither mRNA nor protein expression of TRA2A or TRA2B were able to clearly discriminate *TRA2A*-dependent vs independent lines, we did note that three *TRA2A*-dependent lines (PANC0504, NCI-H23, and LN319) harbored the lowest copy numbers of *TRA2B* (S1G Fig). This suggests that *TRA2A* dependency in some, but not all, cases may arise from partial copy number loss in its paralog partner *TRA2B*, similar to what has been described for other paralog dependencies [18,41,42]. In conclusion, our results demonstrate that *TRA2A* is a strong and selective dependency in a subset of cancer cell lines (Fig 1J).

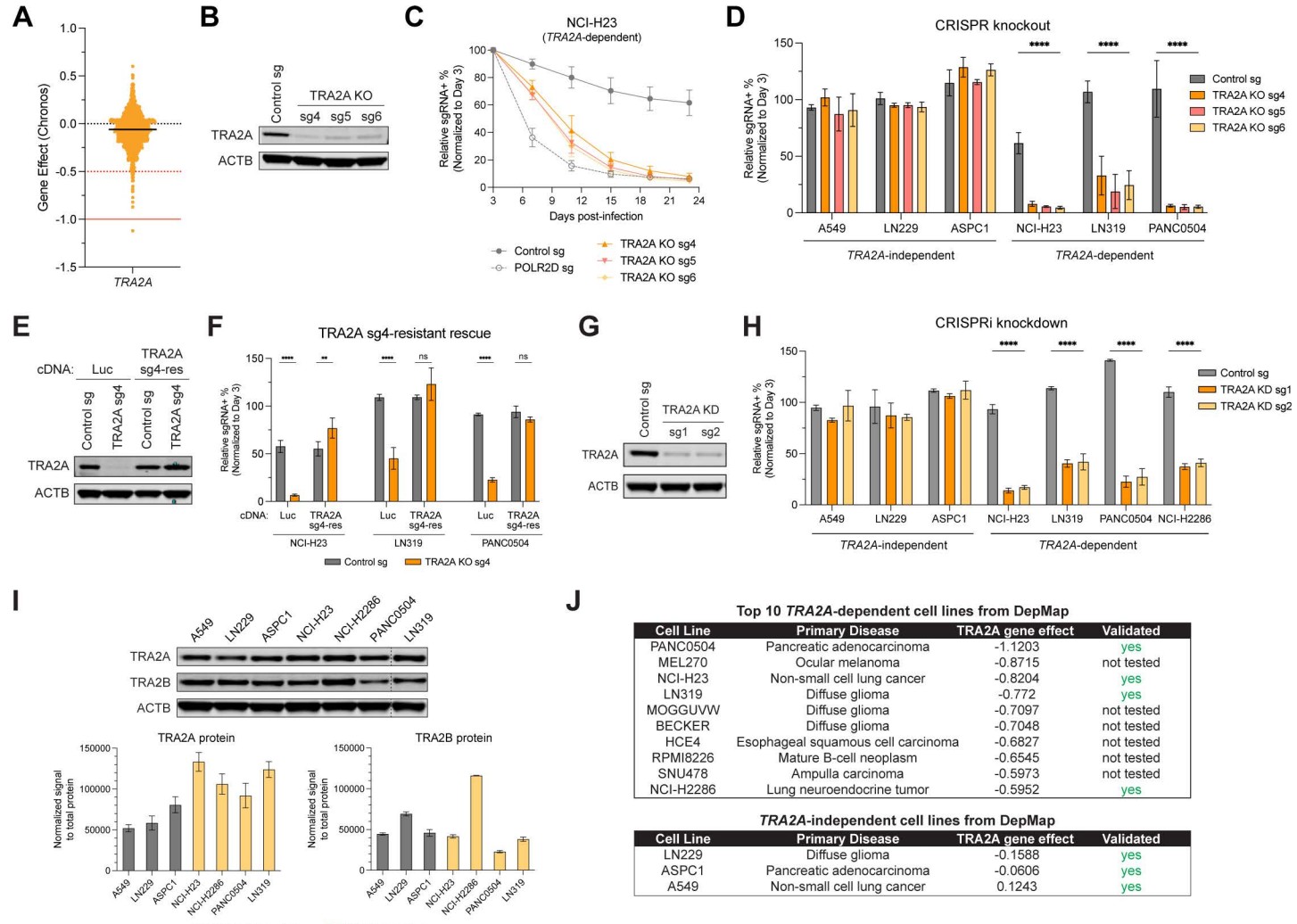

**Fig 1. *TRA2A* is a selective dependency in a subset of cancer cell lines.** (A) Distribution of gene effect scores after *TRA2A* knockout from the Cancer Dependency Map (Public 24Q2 dataset). Red lines mark gene effect scores of -0.5 (dotted) and -1.0 (solid). (B) Immunoblot showing depletion of TRA2A protein levels after CRISPR-Cas9 targeting in NCI-H23 cells. ACTB was used as a loading control. (C,D) Competition-based proliferation assays performed in indicated Cas9+cell lines after *TRA2A* KO. SgRNA+ populations were monitored over time with a co-expressed fluorescent protein marker. Plotted is the relative sgRNA+ population normalized to the Day 3 sgRNA+ population. Bar graph summarizes normalized percentage of sgRNA+ cells remaining at Day 22 or 23 after infection, n = 3. (E) Immunoblot showing depletion or rescue of TRA2A protein levels in NCI-H23 Cas9+cells stably expressing luciferase (control) or *TRA2A* cDNA resistant to sg4 (sg4-res) and infected with control or *TRA2A*-targeting sgRNAs. (F) Competition-based rescue assay performed in indicated *TRA2A*-dependent Cas9+cells also stably expressing luciferase or *TRA2A* cDNA resistant to sg4 (sg4-res) and subsequently infected with control or *TRA2A*-targeting sgRNAs. Shown is the percentage of sgRNA+ cells at Day 28 after infection, normalized to Day 3, n = 3. (G) Immunoblot showing depletion of TRA2A protein levels after CRISPRi-mediated repression in NCI-H23 cells. (H) Competition-based proliferation assays performed in indicated Zim3-dCas9+cell lines after *TRA2A* KD. SgRNA+ populations were monitored over time with a co-expressed fluorescent protein marker. Shown is the percentage of sgRNA+ cells at Day 28 after infection normalized to Day 3, n = 3. (I) Immunoblots showing protein levels of TRA2A, TRA2B, and ACTB in *TRA2A*-independent and -dependent cell lines. Quantification was performed by normalizing protein signal to total protein stain (see S1D Fig), n = 2. (J) Summary of top 10 *TRA2A*-dependent cell lines and 3 tumor type-matched *TRA2A*-independent cell lines from DepMap. (D,F,H,I) Error bars represent standard deviation from the mean. (**)$P < 0.01$, (***)$P < 0.001$, (****)$P < 0.0001$, and (ns) not significant, as calculated by repeated measures two-way ANOVA followed by Dunnett's multiple comparison test.

## Genetic modifier screening identifies paralog synthetic lethality between *TRA2A* and *TRA2B*

To systematically identify genetic modifiers of *TRA2A* dependency, we pursued unbiased genome-wide screens in isogenic *TRA2A* wildtype vs knockout conditions (Fig 2A). We took advantage of a previously described strategy for

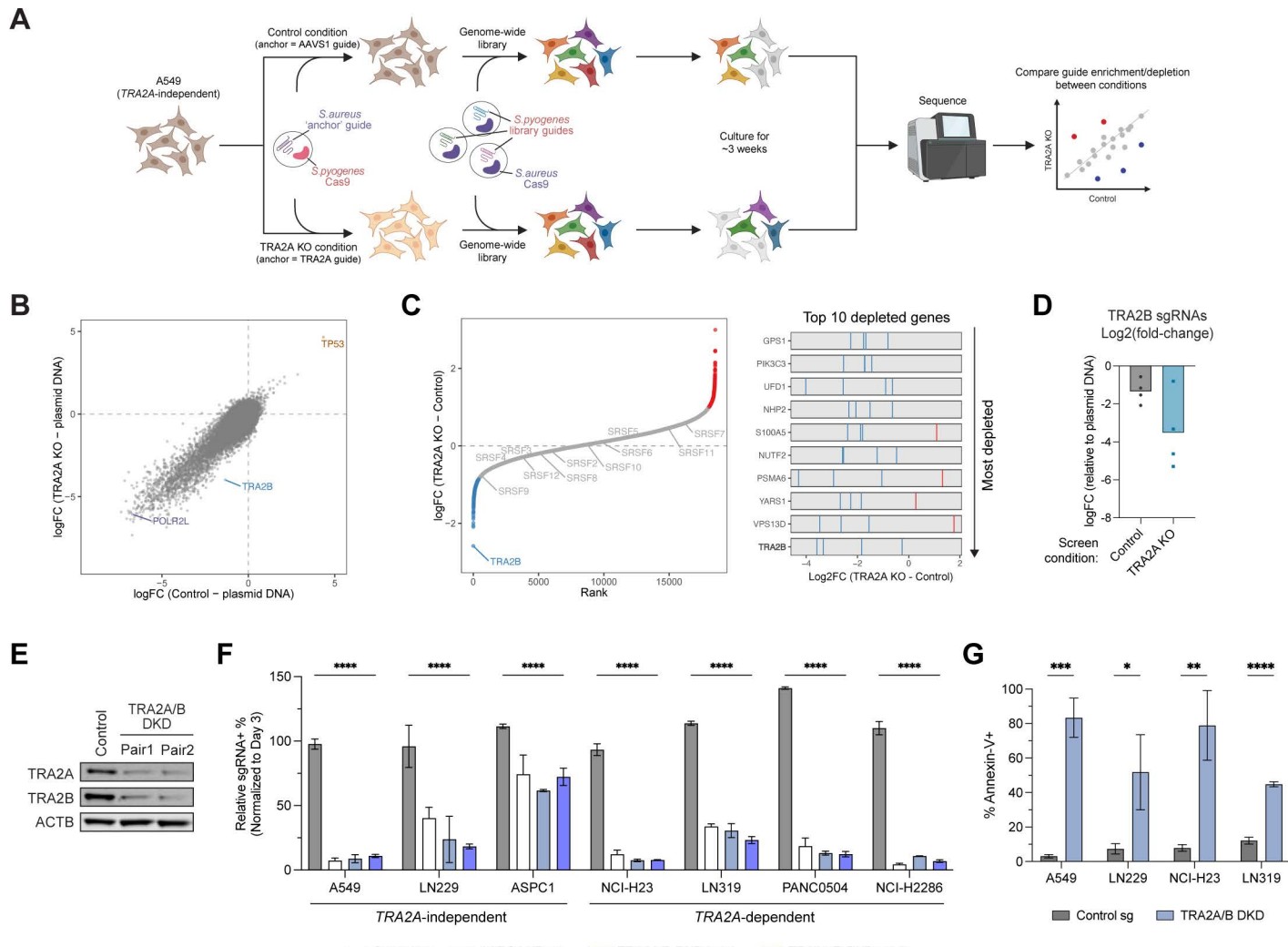

**Fig 2. Genetic modifier screening identifies paralog synthetic lethality between *TRA2A* and *TRA2B*.** (A) Schematic representation of CRISPR screen for genetic modifiers of *TRA2A* dependency. (B) Scatterplot of log2 fold-change (logFC) in A549 cells between control compared to the plasmid DNA library (x-axis) and *TRA2A* KO compared to the plasmid DNA library (y-axis), calculated by MAGeCK. (C) (Left) Distribution of genes ranked by the log2 fold-change (logFC) between *TRA2A* KO and control conditions in A549, calculated by MAGeCK. The blue and red dots indicate two standard deviations from the mean log2 fold-change. (Right) Log2 fold-change between *TRA2A* KO and control conditions of the individual guides targeting genes representing the 10 most depleted genes in A549 cells. (D) Log2 fold-change of *TRA2B* sgRNAs between control or *TRA2A* KO conditions relative to plasmid DNA library. (E) Immunoblot confirming dual depletion of TRA2A and TRA2B protein levels after CRISPRi targeting. ACTB was used as loading control. (F) Dual-guide CRISPRi competition assays in indicated Zim3-dCas9+cell lines, n = 3. Control condition indicates sgRNAs targeting *AAVS1* and *ROSA26* safe harbor loci, *AARS1* KD indicates pan-essential positive control. Values indicate Day 28 percent sgRNA+ population normalized to Day 3 percent sgRNA+ population. Error bars represent standard deviation from the mean. (****)$P < 0.0001$, as calculated by repeated measures two-way ANOVA followed by Dunnett's multiple comparison test. (G) Quantification of percent AnnexinV+ apoptotic cells upon negative control or *TRA2A/B* DKD at Day 7 post-infection as measured by flow cytometry in the indicated cell lines. Error bars represent standard deviation from the mean. (*)$P < 0.05$, (**)$P < 0.01$, (***)$P < 0.001$, and (****)$P < 0.0001$, as calculated by two-tailed unpaired t-test. Schematic in (A) created in BioRender under License: https://BioRender.com/o10ks6p.

modifier screening that utilizes orthogonal CRISPR-Cas9 systems and avoids the need for deriving single-cell knockout clones [43]. First, we stably transduced cells with *S. pyogenes* Cas9 and an *S. aureus* 'anchor guide' targeting *TRA2A* or as a control, the *AAVS1* safe harbor locus. At this stage, no knockout occurs due to the species mismatch between Cas9 and guide RNA. Subsequently, cells are transduced with *S. aureus* Cas9 and the *S. pyogenes* genome-wide guide library, at which point each orthogonal Cas9 is able to pair with its cognate guide to activate targeting by both systems (Fig 2A). Cells were then cultured for approximately 3 weeks before being collected to quantify changes in guide abundance by sequencing. Prior to setting up the screen, we tested several *S. aureus* guides targeting *TRA2A* and selected the most efficient one to use as the 'anchor guide' (sg1, S2A Fig). We then performed the control and *TRA2A* anchor screens in the *TRA2A*-independent cell line A549. Upon completing the screens, we compared the results of the control and *TRA2A* KO conditions to identify genes that confer sensitivity (i.e., are synthetic lethal) to *TRA2A* KO. The two screen conditions were highly correlated (Fig 2B), with the majority of genes having similar fold-changes whether or not *TRA2A* was simultaneously knocked out. However, the paralog partner *TRA2B* emerged as the top synthetic lethal target (Fig 2C) with multiple independent *TRA2B*-targeting guides showing consistently greater depletion in the *TRA2A* KO condition (Fig 2D). Interestingly, other SR proteins were not as significantly depleted or enriched (Figs 2C and S2B), suggesting that synthetic lethality with *TRA2A* in these cells does not extend to other SR protein family members besides *TRA2B*.

Next, we performed validation experiments to confirm the synthetic lethality between *TRA2A* and *TRA2B* identified in our screen. For this, we used dual knockdown (DKD) competition assays across multiple *TRA2A*-dependent and independent cell lines (Figs 2E, 2F and S2D). *TRA2A/B* DKD led to rapid depletion in all cell lines tested, at rates similar to KD of a positive control pan-essential gene (*AARS1*) (Figs 2F and S2D). Notably, the rate of depletion with *TRA2A/B* DKD was higher compared to *TRA2A* KD alone in *TRA2A*-dependent cell lines (S2D Fig). *TRA2A/B* DKD also induced robust levels of apoptosis (Fig 2G), consistent with the strong depletion in the competition assay. Finally, in support of our findings, data from a previous multiplexed CRISPR screen also implicated *TRA2A* and *TRA2B* as synthetic lethal paralogs in PC-9 and HeLa cells (S2C Fig) [27]. Importantly, all the cell lines tested represent a diverse set of cell types and genetic backgrounds, suggesting that *TRA2A* and *TRA2B* are uniformly co-essential. Altogether, our results underscore the strong interdependence of the *TRA2* paralogs in maintaining cell viability, with *TRA2B* playing a critical role in mitigating the sensitivity to *TRA2A* loss.

### *TRA2A* and *TRA2B* function redundantly to maintain widespread constitutive splicing

Given the synthetic lethal interaction between *TRA2A* and *TRA2B*, we sought to better understand the details of their functional relationship. Since both proteins are known splicing factors, we hypothesized that their interdependence might arise from overlapping roles in splicing regulation. To address this, we performed RNA-sequencing following single or dual KD of *TRA2A* and *TRA2B* in both *TRA2A*-independent (A549) and *TRA2A*-dependent (NCI-H23) cell lines. We first analyzed effects on splicing by quantifying local splicing variations (LSVs) using MAJIQ. LSVs consist of the splice junctions that are spliced to or from a particular reference exon, and under this framework, MAJIQ effectively captures both classical binary events (i.e., cassette exons) as well as more complex types of splicing [44,45]. Splicing was quantified as the amount of inclusion or "percent spliced in" (PSI) of a junction, with changes in splicing represented as delta PSI (dPSI).

Our splicing analysis revealed that while *TRA2A* or *TRA2B* single KD resulted in a comparable number of splicing changes, *TRA2A/B* DKD induced considerably more (4.5 - 7 fold greater) than either single KD condition (Fig 3A and 3B and S3 and S4 Tables). The majority of splicing changes were increases in skipping affecting cassette exons (S3A Fig), consistent with the known roles of TRA2 proteins as exonic splicing enhancers [33,35,36]. To a lesser extent, we also detected increases in splicing inclusion upon single or dual KD (Fig 3A and 3B), which may reflect the ability of TRA2 proteins to also act as splicing repressors when bound to intronic sequences [46], or the indirect effects of other splicing factors. In addition, comparative analysis of changing events from each group indicated that the vast majority of *TRA2A/B* DKD-induced splicing changes were unique to this dual depletion condition (84% in A549 and 86% in NCI-H23, S3B Fig).

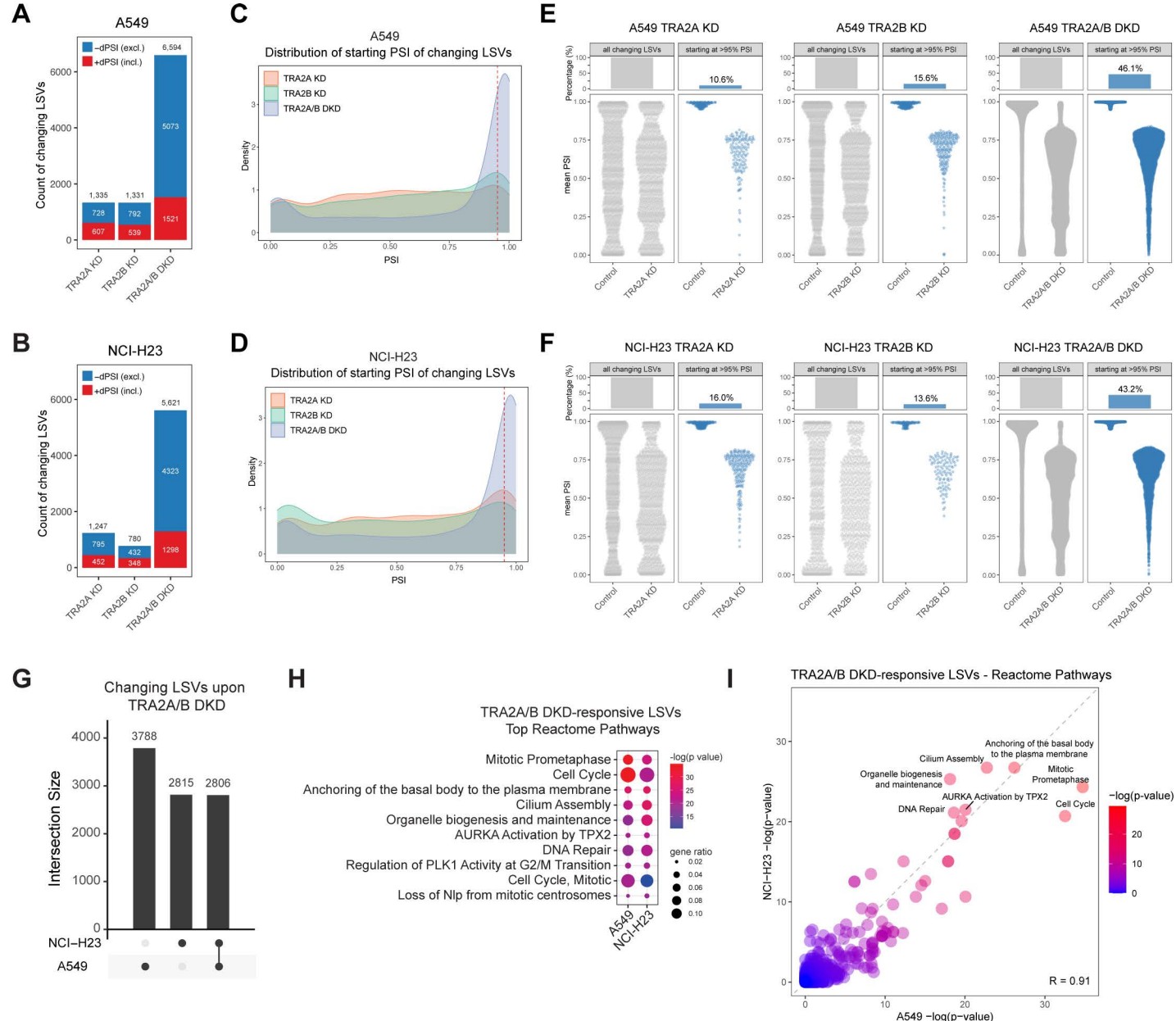

**Fig 3. *TRA2A* and *TRA2B* function redundantly to maintain widespread constitutive splicing.** (A,B) Quantification of number of changing LSVs in (A) A549 and (B) NCI-H23 cells. Changing LSVs were determined by confidence level (probability changing) > 95% and |dPSI| > 10%. Blue represents negative dPSI or LSVs more excluded/repressed upon knockdown. Red represents positive dPSI or LSVs more included/activated upon knockdown. (C,D) Distribution of starting PSI value of changing LSVs in each knockdown condition compared to control in (C) A549 and (*D*) NCI-H23 cells. Red dashed line marks PSI = 95%. (E,F) Mean PSI level for all changing LSVs in control cells and after each knockdown, or subsetted for changing LSVs starting with PSI > 95% in control cells before each knockdown in (E) A549 and (F) NCI-H23 cells. (G) UpSet plot representing the overlap of changing LSVs upon *TRA2A/B* DKD in A549 and NCI-H23 cells. (H) Top enriched Reactome pathways of changing LSVs upon *TRA2A/B* DKD in A549 and NCI-H23 cells. (I) Scatterplot showing enrichment of Reactome pathways for changing LSVs upon *TRA2A/B* DKD in A549 and NCI-H23 cells. Data plotted represents the -log(adjusted p-value) assigned to the Reactome term.

This suggests substantial functional redundancy between the paralogs that only becomes apparent when both are simultaneously depleted. We also identified changes in splicing unique to either *TRA2A* or *TRA2B* single KD (S3B Fig). Approximately 33–50% of changes occurring with single KD were paralog-specific (not shared with KD of other paralog or DKD), suggesting these targets may be preferentially regulated by a single paralog.

Prior work has shown that TRA2 proteins regulate not only alternative splicing, but also appear to be important for constitutive splicing [37], similar to the canonical role of other SR protein family members. To directly assess this, we analyzed the control or 'starting' PSI values of the splicing events changing in each of our knockdown conditions. We found that while single KD-affected events encompassed a broad range of starting PSI values, *TRA2A/B* DKD-affected events were heavily skewed towards constitutive or near constitutive splicing, marked by starting PSI values > 95% (Fig 3C and 3D). The majority of DKD-responsive constitutive junctions in one cell line were constitutively spliced in the other cell line as well (72–76%), supporting their classification as constitutive rather than as cell-line specific alternative splicing events. Nearly half of all DKD-induced splicing changes (43–46%) involved constitutive events, compared to only 11–16% for single KD conditions (Fig 3E and 3F), suggesting *TRA2A* and *TRA2B* play a more extensive role in constitutive splicing than previously appreciated. Additionally, the magnitude of splicing change was particularly pronounced upon DKD, with many constitutive exons showing complete or near-complete skipping (Fig 3E and 3F), indicating some constitutive exons may be entirely dependent on TRA2 protein activity.

Next, to assess the similarity of splicing changes across cell lines, we determined the overlap of *TRA2A/B* DKD changing events between A549 and NCI-H23. We found that approximately half of the splicing changes were shared between the two cell lines, with an overlap of 43% of A549 events (2,806 out of 6,594 total) and 50% of NCI-H23 events (2,806 out of 5,621 total) (Fig 3G). This suggests that, in addition to context-specific functions, *TRA2A* and *TRA2B* jointly regulate a conserved set of splicing targets. This was further supported by pathway analysis of the genes with *TRA2A/B* DKD-affected splicing changes in each cell line. There was a strong enrichment of essential pathways related to the cell cycle, which included regulation of mitosis, primary cilium biogenesis, and DNA repair (Fig 3H). Notably, these enriched pathways were very consistent across both cell lines, with significance values that were highly correlated (Fig 3H and 3I).

Finally, beyond effects on splicing, we also assessed overall gene expression changes across our various conditions. As confirmation of knockdown efficiency, *TRA2A* and *TRA2B* were among the most significantly downregulated genes in their respective KD conditions. Interestingly, for all single KD conditions, there were very few differentially expressed genes (DEGs) (S3C Fig), indicating that effects were occurring primarily at the level of splicing rather than expression. In contrast, *TRA2A/B* DKD resulted in a significantly higher number of DEGs than single KD (S3C Fig) and pathway analysis revealed that downregulated genes were enriched in many of the same cell cycle-related pathways as we had found associated with genes affected by splicing changes (S3D Fig and Fig 3H). Although a minor proportion of all DEGs (19% in A549 and 17% in NCI-H23) were also associated with splicing changes (S3E Fig), this overlap was not statistically significant, suggesting changes in gene expression are not predominantly driven by direct splicing-based regulation. Collectively, our results demonstrate that *TRA2A* and *TRA2B* share extensive functional redundancy and are jointly required for maintaining a large number of constitutive splicing events. Their overlapping targets are enriched for genes involved in critical processes related to cell cycle and DNA repair, providing insight into the underlying basis of their synthetic lethality.

### *TRA2A* dependency is associated with lack of paralog compensation in splicing of cell cycle-related genes

We next analyzed our RNA-seq data with a focus on understanding the mechanisms driving selective dependency on *TRA2A*. We hypothesized that sensitivity or insensitivity to *TRA2A* loss would be reflected in the splicing changes unique to each of these contexts. To investigate this, we started by directly comparing the LSVs responsive to *TRA2A* KD between our *TRA2A*-dependent (NCI-H23) and *TRA2A*-independent (A549) cell lines. We found that while *TRA2A* depletion resulted in a similar number of splicing changes between the two cell lines, only a small fraction of LSVs were overlapping (~8%) and the majority were specific to each cell line (Fig 4A). Furthermore, each of these context-specific

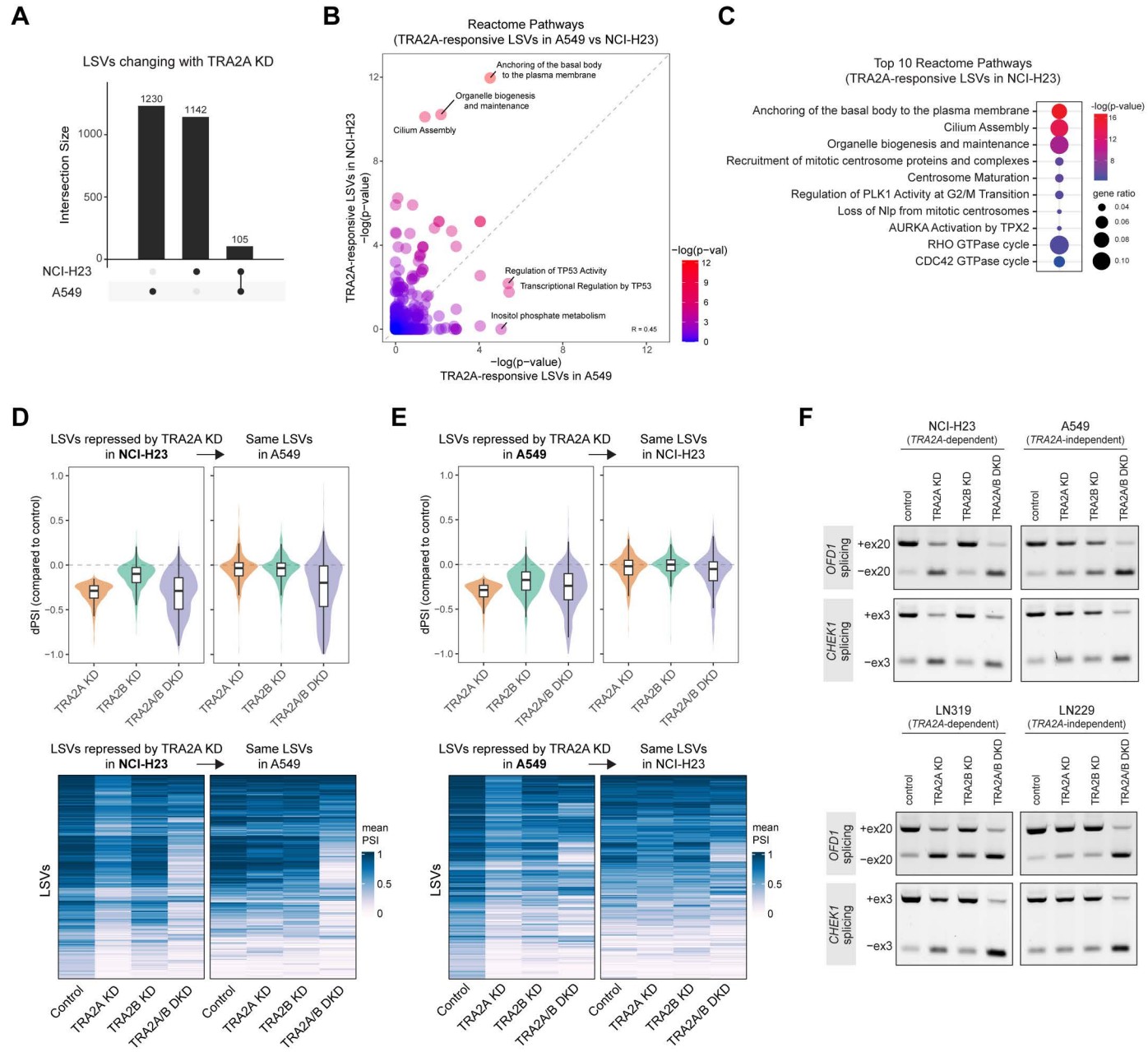

**Fig 4. *TRA2A* dependency is associated with lack of paralog compensation in splicing of cell cycle-related genes.** (A) UpSet plot representing the overlap of changing LSVs upon *TRA2A* KD in NCI-H23 and A549 cells. (B) Scatterplot showing enrichment of Reactome pathways for changing LSVs upon *TRA2A* depletion in A549 and NCI-H23. Data plotted represents the -log(adjusted p-value) assigned to the Reactome term. (C) Top enriched Reactome pathways of changing LSVs responsive to *TRA2A* KD in NCI-H23 cells. (D) (Top) Mean dPSI values of changing LSVs repressed upon *TRA2A* KD in NCI-H23 cells, plotted for both NCI-H23 and A549 cells. (Bottom) Heatmap representation of the same LSVs showing their mean PSI values across all conditions in NCI-H23 and A549. Rows represent an individual LSV and heat color represents the mean PSI for a given LSV, n = 3. (E) (Top) Mean dPSI values of changing LSVs repressed upon *TRA2A* KD in A549 cells, plotted for both A549 and NCI-H23 cells. (Bottom) Heatmap representation of the same LSVs showing their mean PSI values across all conditions in A549 and NCI-H23. Rows represent an individual LSV and heat color represents the mean PSI for a given LSV, n = 3. (F) RT-PCR validation of selected LSVs in *OFD1* and *CHEK1*, performed in indicated *TRA2A*-dependent and independent cell lines.

splicing changes upon *TRA2A* KD were enriched for distinct Reactome pathways (Fig 4B). The most enriched pathways in A549 cells were linked to TP53 activity, while the *TRA2A*-dependent NCI-H23 cell line showed greater enrichment for multiple cell cycle-related pathways (Fig 4C). Most notably, several of the pathways enriched in NCI-H23 cells with *TRA2A* KD were the same ones enriched in both cell lines with *TRA2A/B* DKD (Fig 3H). These included processes related to biogenesis of the primary cilium, an important centrosome-derived organelle with key roles in cell cycle regulation [47].

The commonality in pathway enrichment prompted us to perform an expanded comparison of splicing changes across our various knockdown conditions. We specifically took the LSVs repressed by *TRA2A* KD in NCI-H23 cells and examined these same LSVs in the A549 data (Fig 4D). As expected from the earlier lack of direct overlap, *TRA2A* KD in A549 cells had little effect on these NCI-H23-specific LSVs. However, when both paralogs were depleted with *TRA2A/B* DKD, we saw that the NCI-H23-specific LSVs were now repressed in A549 cells as well (Fig 4D, A549 boxes on right). This suggested that many of the LSVs reliant on TRA2A for splicing in NCI-H23 cells were, in A549 cells, shared by both paralogs and effectively buffered by TRA2B when TRA2A was depleted. This also explained the overlap in enriched pathways we observed between *TRA2A* KD in the NCI-H23 context and *TRA2A/B* DKD in all contexts. Importantly, the reciprocal comparison did not show the same pattern, with A549 *TRA2A* KD-specific LSV changes not mirrored in NCI-H23 cells upon either *TRA2A* KD or *TRA2A/B* DKD (Fig 4E). We also did not observe any differences between contexts when comparing LSVs activated upon *TRA2A* KD (S4A and S4B Fig), indicating these splicing changes are not subject to the same lack of paralog compensation as *TRA2A* KD-repressed events.

We next sought to determine whether these results extended to additional *TRA2A*-dependent contexts. To this end, we performed RNA-seq following *TRA2A* KO in a pair of glioma cell lines that we had validated to be *TRA2A*-dependent and independent (LN319 and LN229, respectively) (S4C–S4E Fig and S5 and S6 Tables). Similar to our earlier results, direct comparison of *TRA2A*-responsive LSVs between these cell lines yielded only a small degree of overlap (S4C Fig) and Reactome pathway analysis also showed considerable differences in enrichment (S4D Fig). While the LSVs from *TRA2A*-independent LN229 cells exhibited no significantly enriched pathways, the LSVs from *TRA2A*-dependent LN319 cells were highly enriched for pathways involving cell cycle regulation and DNA repair (S4E Fig). Many of these same pathways were also enriched in the analysis of *TRA2A/B* DKD in NCI-H23 and A549 cells (Fig 3H), further supporting that TRA2A loss in *TRA2A*-dependent cell lines affects splicing of targets typically shared by both *TRA2A* and *TRA2B*. Finally, we performed RT-PCR validation of candidate splicing events in the cell cycle-related genes *OFD1* and *CHEK1* (Fig 4F). OFD1 is a centrosomal protein important for formation of the primary cilium [48] and CHEK1 is a cell cycle checkpoint protein previously shown to be a downstream target of the TRA2 paralogs [37]. We found that for both of these splicing events, *TRA2A* KD in *TRA2A*-dependent cell lines increased skipping to a similar extent as dual paralog depletion, while *TRA2A* KD in *TRA2A*-independent cell lines showed minimal effects, indicative of effective buffering by *TRA2B* (Fig 4F). Altogether, our results demonstrate that *TRA2A* dependency is associated with a lack of sufficient compensation by *TRA2B*, resulting in mis-splicing of conserved targets involved in essential cell cycle-related processes.

## Loss of *TRA2A* in *TRA2A*-dependent cells results in cell death from defects in mitosis

Our analysis indicated that loss of *TRA2A* in *TRA2A*-dependent cell lines leads to widespread mis-splicing of genes involved in proper assembly of centrosomal complexes and progression through mitosis (Fig 4C). Given that disruption of these processes can trigger mitotic checkpoint activation and cell death [49–51], we next investigated the specific phenotypic consequences of *TRA2A* depletion. For these experiments, we made exclusive use of CRISPRi-mediated KD to avoid the potential complications of gene-targeting with CRISPR KO, which induces DNA double-strand breaks and can increase the incidence of mitotic errors [52]. We first performed cell cycle analysis in both *TRA2A*-dependent and independent cell lines. Following *TRA2A* KD, only *TRA2A*-dependent cell lines exhibited altered cell cycle profiles, with consistent reductions in the G1 population and increases in the G2 population (Figs 5A and S5A). There was also an appreciable

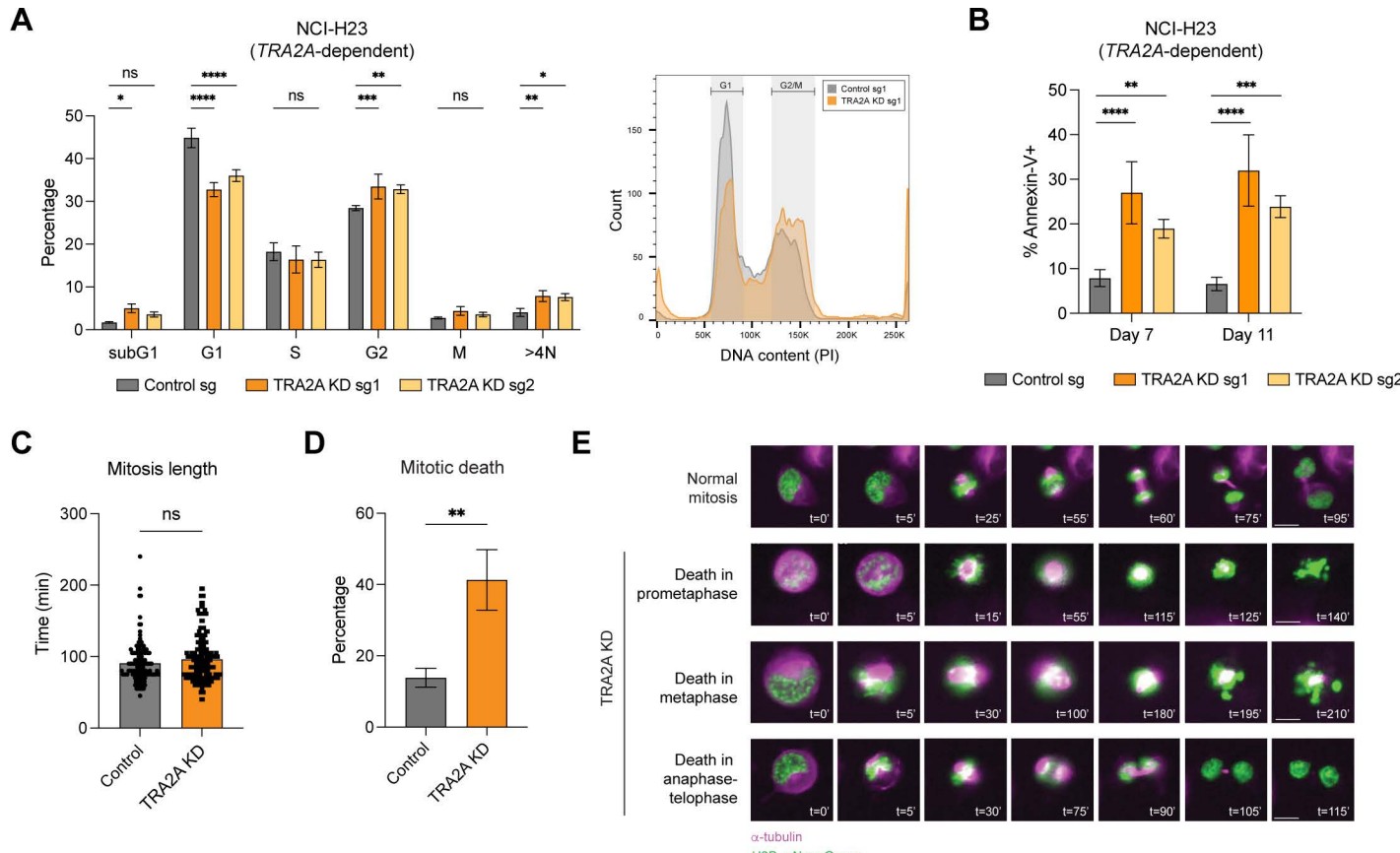

**Fig 5. Loss of *TRA2A* in *TRA2A*-dependent cells results in cell death from defects in mitosis.** (A) (Left) Percentage of cells in each cell cycle stage upon control or *TRA2A* KD in NCI-H23 cells at Day 7 after infection, measured by propidium iodide and phospho-Histone H3 staining followed by flow cytometry, n = 3. Error bars represent standard deviation from the mean. (*)*P* < 0.05, (**)*P* < 0.01, (***)*P* < 0.001, (****)*P* < 0.0001, and (ns) not significant, as calculated by repeated measures two-way ANOVA followed by Dunnett's multiple comparison test. (Right) Representative flow cytometry histogram of propidium iodide staining showing cell cycle distribution upon control and *TRA2A* KD in NCI-H23 cells. (B) Percentage of apoptotic cells upon control or *TRA2A* KD in NCI-H23 at the indicated times after infection, measured by staining for Annexin-V followed by flow cytometry, n = 3. Error bars represent standard deviation from the mean. (**)*P* < 0.01, (***)*P* < 0.001, and (****)*P* < 0.0001, as calculated by repeated measures two-way ANOVA followed by Dunnett's multiple comparison test. (C) Length of mitosis upon control or *TRA2A* KD in NCI-H23 cells, measured by live imaging, n = 3. Each dot represents a cell showing completion of mitosis (control n = 123, *TRA2A* KD n = 209). (ns) not significant, as calculated by two-tailed unpaired t-test. (D) Percentage of mitotic cell population undergoing death during mitosis upon control or *TRA2A* KD in NCI-H23 cells, measured by live cell imaging, n = 3. Error bars represent standard deviation from the mean. (**)*P* < 0.01, as calculated by two-tailed unpaired t-test. (E) Representative images of cells undergoing mitotic death upon control or *TRA2A* KD in NCI-H23 cells stably expressing a chromatin marker (H2B-mNeonGreen; green) and stained with an alpha tubulin dye (magenta). Scale bars = 10 um. For full movies, see S1 Movie.

increase in the fraction of cells with >4N DNA content, which could indicate polyploidy arising from more frequent errors in cell division. *TRA2A* KD also resulted in a pronounced increase in apoptotic cells in *TRA2A*-dependent cell lines, while the viability of non-dependent lines was unaffected (Figs 5B and S5B).

Next, we performed live imaging to monitor progression of cells through the cell cycle. In *TRA2A*-dependent NCI-H23 cells, *TRA2A* KD did not alter the duration of mitosis for cells that successfully completed cell division (Fig 5C), but did significantly increase the amount of cell death occurring during mitosis (Fig 5D). Cells died at various different stages throughout and subsequent to mitosis (Fig 5E and S1 Movie), with most deaths occurring during metaphase (S5C Fig). Our results demonstrate that the splicing defects in *TRA2A*-dependent cells upon loss of *TRA2A* manifest as elevated errors in mitosis and mitotic cell death.

### TRA2A dependency is rescued by overexpression of TRA2B

Like other members of the SR protein family, *TRA2A* and *TRA2B* are subject to auto- and cross-regulation through splicing of their poison exons [37,53–55]. Each factor can promote inclusion of the poison exon found in their reciprocal paralog to induce its downregulation. Consequently, depletion of one paralog alleviates this splicing-based suppression and results in upregulation of the other paralog. To determine the role of this cross-regulatory mechanism in *TRA2A* dependency, we evaluated both expression and splicing of *TRA2A* and *TRA2B* upon depletion of each paralog (S6A–S6D Fig). In agreement with previous observations, loss of one paralog resulted in consistent upregulation of the other paralog, with the magnitude of this effect varying across the cell lines tested and most pronounced at the protein level (S6A–S6C Fig). The upregulation of *TRA2A* or *TRA2B* was also associated with modestly reduced inclusion of their corresponding poison exons (S6D Fig). However, among the cell lines examined, there was no clear correlation between *TRA2A* dependency and the compensatory changes in expression or splicing of *TRA2B*, suggesting other factors are likely responsible for the sensitivity to *TRA2A* loss.

Next, we investigated whether *TRA2A* dependency could instead be due to insufficient basal expression of *TRA2B*. While most examples of paralog synthetic lethality involve complete loss of one paralog member, there are other cases arising from more limited hemizygous or shallow deletion of one paralog [18,41,42]. Consistent with this latter model, our earlier analysis of DepMap data had indicated that several *TRA2A*-dependent cell lines had lower *TRA2B* copy number (S1G Fig). Thus, to determine the importance of *TRA2B* levels on *TRA2A* dependency, we ectopically overexpressed *TRA2B* in two *TRA2A*-dependent cell lines (NCI-H23 and LN319), achieving approximately a 3–4 fold increase in levels of TRA2B protein (Fig 6A). As expected from the known cross-regulation between paralogs, overexpression of *TRA2B* also resulted in strong downregulation of TRA2A (Fig 6A). Despite this concomitant decrease in TRA2A, we found that overexpression of *TRA2B* was sufficient to completely rescue sensitivity to *TRA2A* KO (Fig 6B), suggesting it is the overall dosage of *TRA2A* and *TRA2B* that is essential for cell viability.

To investigate rescue by *TRA2B* at the splicing level, we performed RNA-seq on cells with or without *TRA2B* overexpression (TRA2B-OE) in the context of *TRA2A* KO (S6 Table). Splicing analysis revealed that *TRA2B* overexpression could indeed rescue a large subset of the *TRA2A* KO-responsive splicing events in LN319 cells, with rescue of 344 *TRA2A* KO-repressed events (74% of total, cluster C1) and 49 *TRA2A* KO-activated events (11% of total, cluster C2) (Figs 6C, 6D and S6E). We further validated splicing rescue by performing RT-PCR for splicing events in the cell cycle-related genes *OFD1*, *CHEK1* and *CNTRL* (Fig 6E). Interestingly, a fraction of *TRA2A* KO-responsive events (clusters C3 and C4) showed more limited or lack of rescue by *TRA2B* overexpression (Figs 6C and S6F), indicating that some splicing events may still specifically require *TRA2A* or that the levels of ectopically expressed *TRA2B* were still limiting. Altogether, our data provide further evidence of extensive functional redundancy between *TRA2A* and *TRA2B*, and suggest that compromised paralog buffering from insufficient levels of *TRA2B* is responsible for vulnerability to *TRA2A* loss (Fig 6F).

## Discussion

Paralog-based dependencies have provided both novel therapeutic targets for cancer and insight into the basic functional relationships between paralog partners. Here, we uncover a unique cancer vulnerability involving the paralogous splicing factors *TRA2A* and *TRA2B*. We establish that *TRA2A* and *TRA2B* are synthetic lethal, with their co-essentiality driven by a shared role in widespread alternative and constitutive splicing regulation. While the buffering between these paralogs is intact in most contexts, we discover exceptions where *TRA2A* becomes selectively essential due to insufficient compensation by *TRA2B*.

While our results suggest that *TRA2A* dependency is primarily due to lack of compensation by *TRA2B*, the underlying causes of *TRA2B* insufficiency appear to be complex. Several *TRA2A*-dependent cell lines have lower *TRA2B* copy number (based on data from DepMap), yet this did not always correlate with a proportionally lower amount of TRA2B protein (Figs 1I and S1G). The lack of correlation may reflect the effects of protein dosage compensation, which was recently

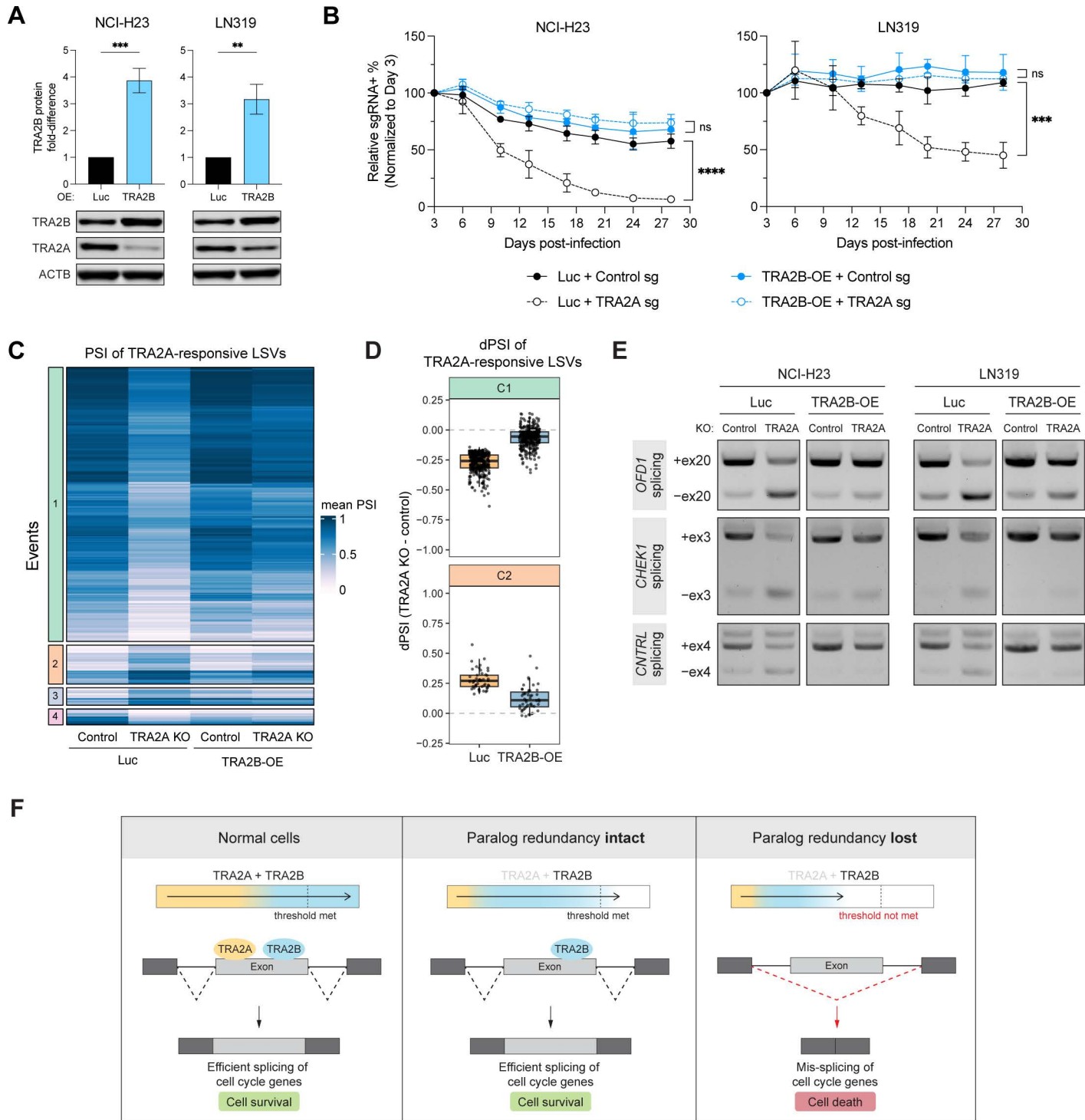

**Fig 6. *TRA2A* dependency is rescued by overexpression of *TRA2B*.** (A) Immunoblot and quantification of TRA2B protein expression relative to ACTB in NCI-H23 and LN319 cells stably expressing luciferase control (Luc) or overexpressing *TRA2B* (TRA2B-OE), n = 3. Error bars represent standard deviation from the mean. (**)$P < 0.01$ and (***)$P < 0.001$, as calculated by two-tailed unpaired t-test. (B) Competition assay performed in NCI-H23 and LN319 Cas9 + cells stably expressing luciferase control (Luc) or overexpressing *TRA2B* (TRA2B-OE) and subsequently infected with control or *TRA2A* sgRNAs, n = 3. Error bars represent standard deviation from the mean. (***)$P < 0.001$, (****)$P < 0.0001$, and (ns) not significant, as calculated by repeated measures two-way ANOVA followed by Tukey's multiple comparison test. (C) Hierarchical clustering of splicing changes detected by MAJIQ in

LN319 cells stably expressing luciferase control (Luc) or overexpressing *TRA2B* (TRA2B-OE) and then infected with control or *TRA2A* sgRNAs (*TRA2A* KO). Rows represent LSVs responsive to *TRA2A* KO compared to control from LN319 cells expressing luciferase. Heat represents the mean PSI for a given LSV, n = 3. (D) Mean dPSI of *TRA2A* KO-responsive LSVs for clusters C1 and C2 in luciferase and TRA2B-OE conditions. (E) RT-PCR validation of candidate *TRA2A* KO-responsive splicing changes that are rescued by *TRA2B* overexpression in NCI-H23 and LN319 cells. (F) Proposed model of TRA2A/TRA2B compensatory mechanisms in normal cells (left), upon TRA2A loss in cells with an intact paralog buffering mechanism (middle), and upon TRA2A loss in cells with a compromised paralog buffering mechanism (right).

reported as a prevalent phenomenon in aneuploid cancer cells [56]. In addition, several other *TRA2A*-dependent cell lines, including one we validated (NCI-H2286), do not exhibit reduced *TRA2B* copy number, suggesting other factors in these cells drive *TRA2A* dependency. Nonetheless, partial copy number loss of *TRA2B* may be sufficient in some cases to make cells *TRA2A*-dependent, similar to other previously described paralog dependencies [18,41,42].

Beyond protein levels, differences in TRA2B protein localization or sequestration may also influence its buffering capacity. Both TRA2A and TRA2B co-localize with RNA inclusions, such as CGG-repeat aggregates, stress granules, and nuclear speckles [57–59]. This sequestration can impair their functions in splicing regulation, highlighting how protein localization directly impacts functional availability. Importantly, these localization patterns can vary significantly across cell types and disease states, and may be contributing to the inability of TRA2B to compensate for *TRA2A* loss in certain cellular contexts. Another potential factor may be the differential expression or availability of important splicing cofactors. While TRA2A and TRA2B share significant overall sequence identity, they are the most divergent in their RS domains, which are critical for facilitating protein-protein interactions [36,60]. Thus, the protein interactome of each paralog is likely to be distinct. If TRA2B-specific cofactors that are critical for its splicing activity become limiting, this may ultimately impair its ability to buffer *TRA2A* loss, even if levels of TRA2B itself remain normal. Further investigation into the specific protein-protein interaction networks of TRA2A and TRA2B will provide valuable insights into the regulators and modifiers of their splicing activity.

Due to the extensive functional redundancy between *TRA2A* and *TRA2B*, single paralog knockdown was unable to detect a substantial fraction of splicing events regulated by these factors. We leveraged the efficient multiplexing capability of CRISPRi [61,62] to simultaneously knockdown both paralogs and reveal the full scope of their collective activity. As the majority of human genes have at least one paralog [63], single gene perturbations likely underestimate the entirety of a gene's functions. Paralog redundancy can also mask functions to such an extent that individual gene knockout results in a complete lack of detectable phenotype [64]. Many splicing factors and other RNA-binding proteins have one or more co-expressed paralogs but have been predominantly studied by single gene knockout/knockdown, such as in several recent large-scale efforts [65,66]. We anticipate that multi-gene perturbation will continue to be a powerful approach for understanding and disentangling paralog functions, especially with the development of higher-order multiplexing methods [28,67,68].

While many functional classes of paralog-based cancer dependencies have been studied thus far, it remains unclear which would serve as the most effective therapeutic targets. Prioritization is especially important as high-throughput multiplexed screening methods have enabled much more efficient and unbiased identification of synthetic lethal paralog pairs [7]. In particular, the strength or 'penetrance' of a given synthetic lethality can be highly variable across different contexts, placing limitations on its potential as a therapeutic strategy [69]. Therefore, synthetic lethal paralogs involved in universally essential cellular processes, such as RNA splicing, may be more likely to provide the ideal 'fully penetrant' phenotype across all contexts. Multiple genome-wide monogenic screens have consistently found that essential genes are highly enriched for those involved in RNA splicing and processing [70–72]. In addition, paralog pairs which exhibited 'symmetric' or reciprocal dependency were also enriched for genes related to RNA splicing [18]. Thus, paralogs involved in the regulation of RNA splicing, such as *TRA2A*/*TRA2B*, may represent a source of particularly strong synthetic lethalities and effective therapeutic targets for cancer.

## Materials and methods

### Cell culture

HEK293T, A549, LN229, and LN319 were cultured in DMEM (Gibco #10569010) supplemented with 10% (v/v) FBS (Thermo Scientific #A3160502) and 1% Penicillin/Streptomycin (Thermo Scientific #15140122). NCI-H23, NCI-H2286 and AsPC-1 (ASPC1) were cultured in RPMI-1640 (Gibco #72400047) supplemented with 10% FBS and 1% Penicillin/Streptomycin. Panc05.04 (PANC0504) were cultured in RPMI-1640 supplemented with 15% FBS and 1% Penicillin/Streptomycin. All cell lines were cultured in 37°C in 5% $CO_2$.

### CRISPR-Cas9 gene knockout or knockdown

Cas9-expressing cancer cell lines were generated by lentiviral transduction with a custom Cas9 expression vector (pLC061) and selection with 10 ug/mL of blasticidin. All Zim3-dCas9-expressing cancer cell lines were generated by lentiviral transduction with pHR-UCOE-EF1a-Zim3-dCas9-P2A-mCherry (Addgene #188766). Cells were then sorted either once or twice for mCherry+ cells using a BD FACSAria Fusion to isolate a high purity Zim3-dCas9 expressing population. Single guide RNA sequences were designed using the CRISPick tool from the Broad Institute (https://portals.broadinstitute.org/gppx/crispick/public) and cloned into lentiviral vectors expressing sgRNAs only, or co-expressing Cas9 or Zim3-dCas9. Dual guide sgRNA vectors were cloned into a lentiviral backbone with one sgRNA driven by a hU6 promoter on the forward strand and a second sgRNA driven by a mU6 promoter on the reverse strand. All sgRNA sequences are provided in S1 Table. For production of lentivirus, 293T cells were transfected with psPAX2 (Addgene #12260), pCMV-VSV-G (Addgene #8454) and the appropriate transfer vector using the CalPhos Mammalian Transfection kit (Takara #631312). Lentivirus was harvested 48 hours after transfection in DMEM media supplemented with 10% FBS and passed through 0.45 um filters before use in transduction of cells. Transduced cells were selected in 2 ug/ml puromycin or 10 ug/ml blasticidin for 4–5 days. Gene knockdown or knockout was confirmed by immunoblotting.

### Competition assay

Single-guide or dual-guide competition assays were performed as follows: sgRNAs were cloned into a lentiviral vector co-expressing an mTagBFP2 fluorescent reporter and transduced into Cas9-expressing or Zim3-dCas9-expressing cell lines at an appropriate viral titer to achieve a~50% BFP+ (sgRNA+) population. BFP measurements were taken by flow cytometry on day 3 post-infection and every 3–4 days thereafter for 28 days in total. For analysis, values were normalized to the % BFP+ on day 3 post-infection.

### Cloning of TRA2A and TRA2B expression constructs

TRA2A and TRA2B cDNA were amplified by PCR from NCI-H23 cDNA and cloned into pDONR223. To create the sgRNA-resistant TRA2A cDNA construct, silent mutations were introduced into the sgRNA target sequence using the Q5 Site-Directed Mutagenesis protocol (NEB #E0554S). All sequences were confirmed by Sanger sequencing. TRA2A and TRA2B cDNAs were then transferred to a lentiviral expression vector by Gateway cloning.

### CRISPR screening for genetic modifiers of TRA2A dependency

Genome-wide CRISPR screening of modifiers was performed as previously described with slight modifications [43]. First, A549 cells were transduced with a modified version of the anchor vector pXPR_213 (Addgene #133456) expressing *S. pyogenes* Cas9 and the *S. aureus* anchor guide (either targeting TRA2A or the safe harbor locus AAVS1 as control). Selection was started 2 days after infection with blasticidin (5 ug/mL) and cells were selected for 14 days. Second, cells were then transduced in two biological replicates with the screen library, cloned into a modified version of library vector pXPR_212 (Addgene #133457). Sufficient numbers of cells were infected to achieve a library representation of at

least 500 cells per sgRNA at a transduction efficiency of approximately 40%. For infections, polybrene (8 ug/mL) was also added. Selection was started 2 days after infection with puromycin (2 ug/mL) and cells were cultured for a total of 22 days. Genomic DNA was isolated using the NucleoSpin Blood XL kit (Takara Bio #740950.50). PCR amplification of sgRNA sequences was carried out with the NEBNext Ultra II Q5 Master Mix (NEB #M0544L) in 100 ul reactions each with 10 ug genomic DNA. After amplification, the reactions were pooled and PCR products were purified by two-sided SPRI purification. Amplicons were quantified with the NEBNext Library Quant kit for Illumina (NEB #E7630L) and assessed by Bioanalyzer (Agilent) before being sequenced on an Illumina NextSeq550 with single-end 75 bp reads. Sequencing data was analyzed using PoolQ v3 (Broad Institute) to obtain sgRNA counts and quality control metrics. SgRNA rank analysis was performed using the Test function of MAGeCK-RRA [73]. Beta scores were calculated using MAGeCK-MLE and were normalized using the median of the beta scores of the core essential genes. Guide and gene-level results are provided in S2 Table.

## Immunoblotting

Cells were lysed in RIPA Lysis and Extraction Buffer (G Biosciences #786–489) supplemented with Halt Protease and Phosphatase Inhibitor Cocktail (Thermo Scientific #78440). Lysates were cleared by centrifugation at 14,000 rpm for 10 min at 4°C. Samples were denatured for 10 min at 70°C in 1x LDS sample buffer (Invitrogen #NP0007) and 100 mM dithiothreitol (Thermo Scientific #A39255). Proteins were separated by SDS-PAGE on 4–12% Bis-Tris NuPAGE gels, transferred by iBlot2 to nitrocellulose membranes and blocked using Intercept (TBS) blocking buffer (Li-Cor #927–60003). Blots were then probed with target-specific primary antibodies and IRDye-conjugated secondary antibodies before being imaged on the Odyssey CLx Imager (Li-Cor). Primary antibodies and dilutions used were: ACTB (Abcepta #AM1829B, 1:2,000), TRA2A (GeneTex #GTX87998, 1:1,000) and TRA2B (Bethyl #A305-011A, 1:5,000). Secondary antibodies and dilutions used were: anti-mouse IgG 680CW (Li-Cor #926–68070, 1:10,000) and anti-rabbit IgG 800CW (Li-Cor #926–32211, 1:10,000).

## RNA-sequencing

For single gene knockout experiments, cell lines were collected 7 days after transduction with lentiviral vectors co-expressing Cas9 and sgRNAs targeting control loci, TRA2A, or TRA2B. For single and double gene knockdown experiments, cell lines stably expressing Zim3-dCas9 were collected 5 days after transduction with lentiviral vectors expressing sgRNAs targeting control loci, TRA2A alone, TRA2B alone, or both TRA2A and TRA2B. Total RNA was extracted from cells using the Maxwell RSC simplyRNA Cells Kit (Promega #AS1390) and the Maxwell RSC48 Instrument (Promega). RNA-seq library preparation and paired-end 150-bp sequencing was performed by Novogene. Briefly, mRNA was purified from total RNA using poly-T oligo-attached magnetic beads. After fragmentation, first strand cDNA was synthesized using random hexamer primers. Then the second strand cDNA was synthesized using dUTP. The directional library was ready after end repair, A-tailing, adapter ligation, size selection, USER enzyme digestion, PCR amplification, and purification. After quality control, libraries were pooled and sequenced on the Illumina platform to a minimum depth of 80–100 million read pairs per library.

## Differential splicing analysis

Reads were aligned to the Gencode v34 annotation (hg38/GRCh38) using STAR v2.7.3 [74]. Alternative splicing events were quantified using the Modeling Alternative Junction Inclusion Quantification (MAJIQ) algorithm v2.5 and VOILA (visualization package) [45], requiring a minimum of 10 reads per sample per junction for quantification. Uniquely mapped, junction-spanning reads were used by MAJIQ to build splice graphs for transcripts with both intron detection on and off. The resulting gene splice graphs were analyzed for all identified local splicing variations (LSVs). For every junction in each LSV, MAJIQ quantified expected percent spliced in (PSI) values in control and knockdown samples and expected change

in PSI (dPSI) between control and knockdown samples. MAJIQ results with intron detection on were analyzed with VOILA modulize to quantify specific classes of splicing events. Significantly changing events were identified with event changing = TRUE, indicating a dPSI value greater than 10% for all junctions within an event. MAJIQ results with intron detection off were analyzed with VOILA tsv to quantify LSVs of exonic splicing events. Significantly changing LSVs were identified with a probability changing value > 0.95 and a |dPSI| value greater than 10%. Gene ontology analysis was performed using the gProfiler2 R package v0.2.3 [75]. Visualization and downstream analyses were conducted in R v4.4.2 using the ggplot2 and tidyverse packages. Results of MAJIQ analysis are provided in S3–S6 Tables.

### Differential gene expression analysis

Transcript levels were quantified with kallisto v0.48.0 [76] using the Gencode v34 annotation (hg38/GRCh38). All subsequent analyses were conducted in R v4.4.2 using Bioconductor v3.19. Differential expression analysis was performed using the limma-voom pipeline [77,78]. First, transcript quantification data were normalized using the trimmed mean of M values (TMM) method in edgeR [79]. Genes with <1 CPM in n + 1 samples, where n is the size of the smallest group of replicates, were filtered out. The mean-variance relationship was estimated using the voom function in limma, and differentially expressed genes were identified with the linear modeling using limma. P values were corrected for multiple hypothesis testing using Benjamini-Hochberg false discovery rate correction to generate adjusted p values.

### Semiquantitative RT-PCR

Reverse transcription was performed from 1ug total RNA in a 20 ul reaction, using LunaScript RT SuperMix (NEB #E3010) following manufacturer's instructions. Splicing changes were assessed by PCR using primers in exons flanking the alternatively spliced exon. For each PCR reaction, 1ul cDNA (diluted 1:5) was used as a template in a 20 ul PCR reaction using Q5 Hot Start High Fidelity DNA polymerase (NEB #M0493L). The amplified products were separated by electrophoresis in a 1.5% agarose gel in 1X lithium boric acid buffer (Faster Better Media #LB10–1) stained with SYBR Safe (Thermo #S33102). All primers used for splicing assays are provided in S1 Table.

### Cell cycle assay

Cells were trypsinized, fixed in pre-chilled 70% ethanol in PBS and stored at -20°C overnight. The next day, cells were centrifuged, washed, and stained with phospho-Histone H3 (Ser10) antibody conjugated to Alexa Fluor 647 (Cell Signaling Technology #9716) for 1hr. Cells were then washed with Cell Staining Buffer (BioLegend #420201) and incubated with 50 ug/ml propidium iodide (Invitrogen #P3566) with 0.5 ug/ml RNase A (Thermo Scientific #EN0531) in PBS for 30min. Cells were assessed by flow cytometry and data was analyzed using FlowJo v10.

### Apoptosis assay

Cells were trypsinized, washed with Cell Staining Buffer (BioLegend #420201) and stained with FITC-conjugated Annexin V and 7-aminoactinomycin D (7-AAD) according to manufacturer's instructions (BioLegend #640922). Cells were assessed by flow cytometry and data was analyzed using FlowJo v10.

### Live-cell imaging of mitosis

NCI-H23 cells were transduced with a lentiviral vector expressing H2B-mNeonGreen (H2B-mNG) and selected in hygromycin (100 ug/ml) for 14 days. Cells stably expressing H2B-mNG were then transduced with a lentiviral vector co-expressing Zim3-dCas9 and sgRNA, and selected in puromycin (2 ug/ml) starting 24 hours post-infection. At 96 hours post-infection, cells were seeded at 30K cells/well in a 24-well black-walled, glass-bottom plate, to achieve ~30% confluency the next day. To enrich for cells undergoing mitosis during imaging, cells were synchronized by treatment with 5mM thymidine (Sigma-Aldrich #T1895-1G) for 24 hours and then released into complete growth media. 4 hours before

imaging, cells were incubated in media containing SPY650-tubulin dye (Cytoskeleton #CY-SC503, 1:2000) to visualize alpha-tubulin, and retained in SPY650-tubulin-containing media during imaging. Cells were imaged every 5 minutes on the Zeiss Axio Observer 7 widefield microscope with 20X objective. All images were processed using FIJI software (v2.14.0).

## Supporting information

**S1 Fig. _TRA2A_ is a selective dependency in a subset of cancer cell lines.** (A) Distribution of primary diseases among cancer cell lines from the Cancer Dependency Map (Public 24Q2 dataset) with a _TRA2A_ gene effect score of less than -0.5. (B) Distribution of gene effect scores after knockout of indicated SR protein genes from the Cancer Dependency Map (Public 24Q2 dataset). Red lines mark gene effect scores of -0.5 (dotted) and -1.0 (solid). Also indicated are genes designated as 'common essential' by DepMap. (C) Competition-based proliferation assays performed in indicated _TRA2A_-independent and dependent Cas9+ cell lines after _TRA2A_ KO, n = 3. (D) Representative image of total protein stain for normalization of TRA2A and TRA2B protein levels across cell lines (see Fig 1I). (E-J) Scatterplots from DepMap data of _TRA2A_ gene effect scores vs (E) _TRA2A_ mRNA expression (log2(TPM + 1)), (F) _TRA2B_ mRNA expression (log2(TPM + 1)), (G) _TRA2B_ copy number (log2(relative to ploidy+1)), (H) TRA2A protein expression (z-score), (I) TRA2B protein expression (z-score), or (J) _TRA2B_ gene effect scores. Orange circles indicate cell lines with _TRA2A_ gene effect scores < -0.5, red circles indicate validated _TRA2A_-dependent cell lines and blue circles indicate validated _TRA2A_-independent cell lines. (TIF)

**S2 Fig. Genetic modifier screening identifies paralog synthetic lethality between _TRA2A_ and _TRA2B_.** (A) Immunoblot showing depletion of TRA2A protein levels after _S. aureus_ Cas9 targeting in NCI-H23 cells. ACTB was used as loading control. Bold indicates the guide used in the screen (sg1). (B) Log2 fold-change between _TRA2A_ KO and control conditions of the individual guides targeting genes representing the 10 most enriched genes in A549 cells. (C) CRISPR scores calculated from single and double KO paralog screens in PC-9 and HeLa cells, obtained from Parrish et al. 2021 [27]. (D) Competition assays performed with single or dual KD of _TRA2A_ and _TRA2B_ in indicated Zim3-dCas9+ cell lines, n = 3. (TIF)

**S3 Fig. _TRA2A_ and _TRA2B_ function redundantly to maintain widespread constitutive splicing.** (A) Number of differentially spliced events categorized by event type as detected by VOILA modulizer in A549 and NCI-H23 cells with _TRA2A_ KD, _TRA2B_ KD, or _TRA2A/B_ DKD compared to control. (B) UpSet plots representing the overlap of changing LSVs upon _TRA2A_ KD, _TRA2B_ KD, and _TRA2A/B_ DKD in A549 cells and NCI-H23 cells. (C) Volcano plots for (top) A549 and (bottom) NCI-H23 cells showing the relative fold change in gene expression versus -log(adj. p-values) in each knockdown condition compared to control, calculated by the limma-voom pipeline. Shown on right is a summary of number differentially expressed genes, as determined by an absolute log2FC > 1 and an adjusted p-value < 0.05, n = 3. Blue circles represent downregulated genes, red circles represent upregulated genes and grey circles represent genes not differentially expressed. (D) Top enriched Reactome pathways of (left) downregulated genes and (right) upregulated genes upon _TRA2A/B_ DKD compared to control in A549 and NCI-H23 cells. (E) Venn diagram of overlap between genes showing downregulation, upregulation, and differential splicing upon _TRA2A/B_ DKD in (top) A549 and (bottom) NCI-H23 cells. (TIF)

**S4 Fig. _TRA2A_ dependency is associated with lack of paralog compensation in splicing of cell cycle-related genes.** (A) (Top) Mean dPSI values of changing LSVs activated upon _TRA2A_ KD in NCI-H23 cells, plotted for both NCI-H23 and A549 cells. (Bottom) Heatmap representation of the same LSVs showing their mean PSI values across all conditions in NCI-H23 and A549. Rows represent an individual LSV and heat color represents the mean PSI for a given LSV, n = 3. (B) (Top) Mean dPSI values of changing LSVs activated upon _TRA2A_ KD in A549 cells, plotted for both A549

and NCI-H23 cells. (Bottom) Heatmap representation of the same LSVs from (Top) showing their mean PSI values across all conditions in A549 and NCI-H23. Rows represent an individual LSV and heat color represents the mean PSI for a given LSV, n = 3. (C) UpSet plot representing the overlap of changing LSVs upon *TRA2A* KO in LN229 and LN319. (D) Scatter-plot showing enrichment of Reactome pathways for changing LSVs upon *TRA2A* KO in LN229 and LN319. Data plotted represents the -log(adjusted p-value) assigned to the Reactome term. (E) Top enriched Reactome pathways of changing LSVs responsive to *TRA2A* KO in LN319 cells.
(TIF)

**S5 Fig.  Loss of *TRA2A* in *TRA2A*-dependent cells results in cell death from defects in mitosis.** (A) Percentage of cells in each cell cycle stage upon control or *TRA2A* KD at Day 7 or 11 after infection in indicated cell lines, measured by propidium iodide and phospho-Histone H3 staining followed by flow cytometry, n = 3. (B) Percentage of apoptotic cells upon control or *TRA2A* KD at Day 7 or 11 after infection in indicated cell lines, measured by Annexin-V staining followed by flow cytometry, n = 3. (C) Percentage of dying mitotic cell population in each stage of mitosis during death in NCI-H23 cells, measured by live cell imaging, n = 3. Error bars represent standard deviation from the mean. (*)$P < 0.05$ and (ns) not significant, as calculated by repeated measures two-way ANOVA followed by Dunnett's multiple comparison test. For (A,B) error bars represent standard deviation from the mean. (*)$P < 0.05$, (****)$P < 0.0001$, and (ns) not significant, as calculated by repeated measures two-way ANOVA followed by Dunnett's multiple comparison test.
(TIF)

**S6 Fig.  *TRA2A* dependency is rescued by overexpression of *TRA2B*.** (A) Representative immunoblots measuring protein expression of TRA2A and TRA2B upon knockdown of *TRA2A* or *TRA2B*. ACTB was used as loading control. (B) Quantification of TRA2A protein upon *TRA2B* depletion (left) and TRA2B protein upon *TRA2A* depletion (right) relative to ACTB as in (A), n = 3. Error bars represent standard deviation from the mean. (C) RNA expression of *TRA2A* upon *TRA2B* depletion (left) and *TRA2B* upon *TRA2A* depletion (right) in indicated cell lines, as measured by RNA-seq. Error bars represent standard deviation from the mean. (D) Splicing inclusion of *TRA2A* poison exon upon *TRA2B* depletion (left) and *TRA2B* poison exon upon *TRA2A* depletion (right) in indicated cell lines, measured by RNA-seq and quantified by MAJIQ. Error bars represent standard deviation from the mean. (E) Number of LSVs in each cluster of the heatmap in Fig 6C. (F) Mean dPSI of *TRA2A* KO-responsive LSVs for clusters C3 and C4 in luciferase and TRA2B-OE conditions.
(TIF)

**S1 Table.  Sequences of CRISPR guides and RT-PCR primers.**
(XLSX)

**S2 Table.  Analysis of CRISPR screen in A549 cell line using MAGeCK RRA.**
(XLSX)

**S3 Table.  Differential splicing analysis using MAJIQ of *TRA2A* KD, *TRA2B* KD, and *TRA2A/B* dual KD in A549 cell line.**
(XLSX)

**S4 Table.  Differential splicing analysis using MAJIQ of *TRA2A* KD, *TRA2B* KD, and *TRA2A/B* dual KD in NCI-H23 cell line.**
(XLSX)

**S5 Table.  Differential splicing analysis using MAJIQ of *TRA2A* KO or *TRA2B* KO in luciferase-expressing or *TRA2B*-overexpressing cells in LN229 cell line.**
(XLSX)

**S6 Table. Differential splicing analysis using MAJIQ of *TRA2A* KO or *TRA2B* KO in luciferase-expressing or *TRA2B*-overexpressing cells in LN319 cell line.**
(XLSX)

**S1 Movie. Live imaging of cell death during mitosis in NCI-H23.** (Top left) Control mitotic cell. (Top right) Mitotic cell dying in prometaphase. (Bottom left) Mitotic cell dying in metaphase. (Bottom right) Mitotic cell dying in telophase. Time stamp corresponds to the start of mitosis.
(MP4)

## Acknowledgments

We thank the members of the Choi laboratory for helpful discussions. We also thank Drs. Andrei Thomas-Tikhonenko, Yoseph Barash, Lan Lin, and Kathryn Hamilton for valuable feedback.

## Author contributions

**Conceptualization:** Amanda R Lee, Peter S. Choi.

**Formal analysis:** Amanda R Lee.

**Funding acquisition:** Peter S. Choi.

**Investigation:** Amanda R Lee, Anna Tangiyan, Isha Singh, Peter S. Choi.

**Methodology:** Amanda R Lee, Peter S. Choi.

**Project administration:** Amanda R Lee.

**Supervision:** Peter S. Choi.

**Visualization:** Amanda R Lee.

**Writing – original draft:** Amanda R Lee, Peter S. Choi.

**Writing – review & editing:** Amanda R Lee, Peter S. Choi.

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
