## [Decision Letter · Decision Letter 0]

10 Apr 2025

Dear Dr Choi,

We are pleased to inform you that your manuscript entitled "Incomplete paralog compensation generates selective dependency on TRA2A in cancer" has been editorially accepted for publication in PLOS Genetics. Congratulations!

Yours sincerely,

Shuguo Sun

Guest Editor

PLOS Genetics

Hongbin Ji

Section Editor

PLOS Genetics

Aimée Dudley

Editor-in-Chief

PLOS Genetics

Anne Goriely

Editor-in-Chief

PLOS Genetics

Comments from the reviewers (if applicable):

Reviewer's Responses to Questions

**Comments to the Authors:**

Reviewer #1: Lee et al

During evolution, many genes underwent duplication events. While some duplicates were lost, others persisted as paralogs, often retaining similar functions but also diverging in key ways. This concept is intriguing not only from a therapeutic perspective, where paralog dependencies can be exploited in cancer, but also from a mechanistic standpoint, as it allows us to examine both the similarities and differences between these gene pairs. Therefore, I find the Lee et al manuscript compelling, not only in its focus on splicing regulation but also in its broader exploration of paralog relationships and their biological significance.

The manuscript uses multi-gene perturbation, which is a powerful approach to understanding paralog function.

The logical flow of ideas from paralog redundancy to TRA2A dependency in cancer is clear and well-organized. Each result builds upon the previous findings, leading to well-supported conclusions. Having said that, the manuscript feels a bit dense and would benefit from schematic summaries of the redundancy between the paralogs.

Main points:

• Fig. 3C: Lee et al. identify TRA2 proteins as regulators of constitutive splicing, which is an interesting and novel finding. However, the criteria for defining a splicing event as 'constitutive' are not entirely clear. The study uses RNA-seq across multiple cell lines, yet constitutive splicing is determined based on >95% exon inclusion in a single cell line. This approach could be refined to ensure a more robust classification, potentially by incorporating broader cross-cell line analyses. This concern arises from the fact that an exon can be fully included (100%) in one cell line but completely excluded in another. In such cases, this reviewer would classify the exon as alternatively spliced rather than constitutive.

• The authors state that only a minor proportion of differentially expressed genes (DEGs) (19% in A549 and 17% in NCI-H23) were also associated with splicing changes (Supp. Fig. 3E). However, this percentage does not seem negligible, and depending on statistical analyses, it could be considered significant. It would be helpful to provide statistical testing (e.g., enrichment analysis or permutation testing) to determine whether this overlap is greater than expected by chance. Additionally, further discussion on whether these splicing-associated DEGs are functionally related or enriched in specific pathways would strengthen the interpretation of these results.

• The manuscript demonstrates that TRA2 proteins are important for the splicing of cell cycle genes and for proper cell cycle progression. It would be interesting to explore, through literature or web-based tools (such as https://www.phosphosite.org/), whether these proteins undergo post-translational modifications in a cell cycle-dependent manner. Pointing at such possible modification can add a layer of understanding to the potential regulatory mechanisms linking TRA2 function to cell cycle control.

• GEO accession is pending—this should be completed before final submission.

Reviewer #2: Lee and colleagues characterise a buffering relationship between the paralogous splicing factors TRA2A and TRA2B. TRA2A is identified as being essential in a subset of cancer cell lines in the publicly available DepMap dataset and this is validated by the authors in additional CRISPR experiments. The authors perform a CRISPR screen to identify synthetic lethal partners of TRA2A and identify TRA2B as the top hit. This suggests a negative genetic interaction between TRA2A and TRA2B – loss of both causes a significant growth defect multiple cancer cell lines – but TRA2B mRNA or protein expression on its own is not sufficient to explain TRA2A essentiality (the two are only weakly correlated). The authors then profile splicing in the presence of TRA2A perturbation, TRA2B perturbation, and dual TRA2A/B perturbation and demonstrate that the two paralogs regulate many of the same targets. They suggest that TRA2A is essential in cell lines where TRA2B expression is insufficient to compensate for the splicing defects induced by TRA2A loss (in particular impacting cell cycle genes) and demonstrate that increasing TRA2B expression can rescue the resulting fitness defect.

The study is well written, the results are clear, and the conclusions drawn are consistent with the data presented. It is curious that TRA2B expression on its own is not predictive of TRA2A essentiality across the DepMap cell lines, but the authors acknowledge this and acknowledge that the current results do not fully explain the dependency.

I have one minor suggestions for analyses that might shed light on the question – there are two proteomic profiles of the DepMap Cell Lines (Goncalves et al, Cancer Cell; Nusinow et al Cell). Rather than just looking at the association between essentiality and mRNA abundance (Supplemental Figure 1EFG) it would be worth looking at protein abundance of TRA2B and its association with TRA2B essentiality. It would also be worth adding the Pearson correlation directly to these figures (S1E, S1F, S1G).

Reviewer #3: This manuscript from the Choi lab validates a synthetic lethal interaction between the splicing regulatory factors TRA2A and TRA2B. By leveraging DepMap data, the authors observed that TRA2A exhibits a highly selective cell line dependency. Approximately 10% of TRA2A-dependent cell lines have a low copy number of TRA2B, a functional paralog of TRA2A. Through an elegant anchor CRISPR screen in the TRA2A-independent cell line A549, the authors identify TRA2B as the strongest synthetic lethal interactor of TRA2A, a finding they extensively validate. Through in-depth transcriptomic analyses, the authors demonstrate that co-depletion of TRA2A and TRA2B primarily affects highly included exons in genes associated with the cell cycle. Consistently, they show that TRA2A knockdown in TRA2A-dependent cell lines results in cell cycle defects leading to apoptosis. Finally, the authors demonstrate that TRA2B overexpression rescues TRA2A dependency in TRA2A-sensitive cell lines and reverses the associated splicing changes.

This is a highly compelling study that provides deep mechanistic insight into the synthetic lethal relationship between TRA2A and TRA2B. A limitation of the work is the lack of clarity regarding the mechanisms underlying TRA2A dependency in sensitive cell lines that do not exhibit lower TRA2B copy numbers. The authors propose reasonable hypotheses in the discussion section, and while further investigation into this question could be informative, it is likely beyond the scope of this publication in PLOS Genetics. Overall, this is an extremely well-written manuscript with research conducted at a high level of rigor. As such, I enthusiastically recommend its publication in PLOS Genetics.

Below are a few minor comments for the authors’ consideration:

1. Based on DepMap data, how does TRA2B sensitivity (gene effect) differ between TRA2A-dependent and TRA2A-independent cell lines?

2. It would be helpful to visualize the top 10 buffering gene interactions identified in the CRISPR screen. Perhaps these could be presented as a new panel similar to Figure 2C in Supplementary Figure 2.

3. Have the authors examined whether the novel synthetic lethal interactions of TRA2A (beyond TRA2B) correlate with copy number variations in TRA2A-sensitive cell lines?

4. The study presents the intriguing finding that, in contrast to single knockdowns, co-depletion of TRA2A and TRA2B leads to widespread gene expression changes across all tested cell lines. These changes appear largely independent of the genes affected at the splicing level. It would be valuable for the authors to discuss this observation further. Do they speculate that the TRA2 family plays an underappreciated role in gene expression or RNA decay? Alternatively, could the co-depletion of TRA2A/B activate splicing events that trigger transcript degradation via nonsense-mediated decay (NMD), with these unstable transcripts escaping detection in RNA-Seq? Have the authors performed RNA-Seq experiments in the presence of NMD inhibitors to investigate this possibility?

**Have all data underlying the figures and results presented in the manuscript been provided?**

Reviewer #1: Yes

Reviewer #2: Yes

Reviewer #3: Yes

PLOS authors have the option to publish the peer review history of their article (what does this mean? ). If published, this will include your full peer review and any attached files.

**Do you want your identity to be public for this peer review?** For information about this choice, including consent withdrawal, please see our Privacy Policy .

Reviewer #1: **Yes: ** Maayan Salton

Reviewer #2: No

Reviewer #3: No

**Data Deposition**

http://datadryad.org/submit?journalID=pgenetics&manu=PGENETICS-D-25-00287

**Press Queries**

---

## [Editor Report · Acceptance letter]

PGENETICS-D-25-00287

Incomplete paralog compensation generates selective dependency on TRA2A in cancer

Dear Dr Choi,

We are pleased to inform you that your manuscript entitled "Incomplete paralog compensation generates selective dependency on TRA2A in cancer" has been formally accepted for publication in PLOS Genetics! Your manuscript is now with our production department and you will be notified of the publication date in due course.

With kind regards,

Anita Estes

PLOS Genetics

On behalf of:
